# FARSE-CNN: Fully Asynchronous, Recurrent and Sparse Event-Based CNN

## Abstract

Event cameras are neuromorphic image sensors that respond to per-pixel brightness changes, producing a stream of asynchronous and spatially sparse events. Currently, the most successful algorithms for event cameras convert batches of events into dense image-like representations that are synchronously processed by deep learning models of frame-based computer vision. These methods discard the inherent properties of events, leading to high latency and computational costs. Following a recent line of works, we propose a model for efficient asynchronous event processing that exploits sparsity. We design the Fully Asynchronous, Recurrent and Sparse Event-Based CNN (FARSE-CNN), a novel multi-layered architecture which combines the mechanisms of recurrent and convolutional neural networks. To build efficient deep networks, we propose compression modules that allow to learn hierarchical features both in space and time. We theoretically derive the complexity of all components in our architecture, and experimentally validate our method on tasks for object recognition, object detection and gesture recognition. FARSE-CNN achieves similar or better performance than the state-of-the-art among asynchronous methods, with low computational complexity and without relying on a fixed-length history of events.

## 1 Introduction

Event cameras, or neuromorphic cameras, are image sensors inspired by the biological retina (Posch et al., 2014; Lichtsteiner et al., 2008). While conventional cameras capture snapshots at a predetermined frame rate, event cameras have independent pixels that asynchronously respond to changes of brightness in the scene. When the change crosses a certain threshold, the pixel triggers an event encoding the sign of the measured variation, as in brightness increase or brightness decrease. Thus, an event camera generates a stream of events in space and time. Compared to conventional cameras, event cameras have many advantages that bring them closer to biological vision. Since pixels monitor only brightness changes independently, and they do not have to wait for a global exposure time, events are generated with very high temporal resolution and transmitted by the event camera with low latency, both in the order of microseconds. Additionally, since the photoreceptors work in logarithmic scale, event cameras have a high dynamic range, which exceeds 120 dB. Because event cameras produce data only for pixels where a change is measured, and only at times in which the change occurs, they naturally suppress redundant visual information and the bandwidth is greatly reduced, leading to low power consumption. These properties make event cameras ideal for various applications, such as robotics, autonomous vehicles, IoT and edge computing applications.

To take advantage of these sensors, however, suitable processing techniques and algorithms must be identified. The deep learning methods that are commonly used in conventional computer vision, specifically Convolutional Neural Networks (CNN), cannot be directly applied to the output of event cameras, being it inherently sparse and asynchronous. For this reason, researchers have focused on ways to adapt the event data to the requirements of CNNs, by converting batches of events into synchronous and dense, image-like representations. Despite being effective, this approach discards the inherent properties of event data, and trades off latency and computational costs for task performance. Following a recent trend in event-based vision, we address the problem of developing deep learning techniques that exploit the asynchronous and sparse nature of event data, to allow for efficient event-by-event processing. Previous works have adopted solutions based on 3D point-cloud processing (Sekikawa et al., 2019), graph neural networks (Li et al., 2021; Schaefer et al., 2022),

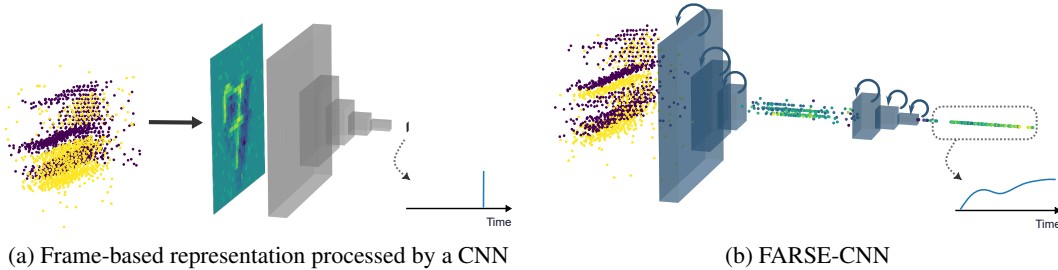

(a) Frame-based representation processed by a CNN          (b) FARSE-CNN

Figure 1: **Overview of FARSE-CNN** in contrast to frame-based event processing. (a) A common approach builds dense event frames that are synchronously processed by a CNN. (b) Convolutional recurrent modules in our architecture, instead, process events directly preserving their spatio-temporal sparsity, and the overall network produces an asynchronous stream of outputs.

sparse convolutions (Messikommer et al., 2020), and memory-augmented networks (Kamal et al., 2023). Inspired by techniques that combine the mechanism of 2D convolution with Recurrent Neural Networks (RNN) (Shi et al., 2015; Cannici et al., 2020), we propose the Fully Asynchronous, Recurrent and Sparse Event-Based CNN (FARSE-CNN), a novel deep architecture that uses recurrent units in a convolutional way to process events in a fully sparse and asynchronous fashion. In order to do this, we design an asynchronous convolutional RNN model based on the Long-Short-Term-Memory (LSTM) (Hochreiter & Schmidhuber, 1997) and a submanifold convolutional RNN variant inspired by works on sparse convolutions (Graham et al., 2018), as well as modules for both spatial and temporal compression of event features. We show that our architecture can be applied to different tasks in event-based vision, achieving performance and computational costs that are competitive to other state-of-the-art methods. Our contributions are summarized as follows:

- We propose FARSE-CNN, a convolutional RNN architecture tailored for asynchronous and sparse event processing

- We propose a novel temporal compression mechanism, in the form of a new Temporal Dropout module, as well as sparse implementations of pooling operations. Our Temporal Dropout introduces hierarchical temporal learning in our models. To the best of our knowledge, this is the first work to explore Hierarchical RNNs (HRNNs) in event vision.

- We derive theoretical bounds that allow to analyze the computational complexity of all elements in our architecture, in terms of floating point operations per event (FLOPs/ev).

- We experimentally demonstrate our method on publicly available datasets for object recognition, gesture recognition and object detection. Our method shows similar performance to the state of the art for asynchronous techniques in object recognition, and improves on it in object detection.

## 2 RELATED WORK

Motivated by the success of deep learning architectures in frame-based computer vision, a number of works in event vision have focused on converting batches of events into dense representations which can be processed using convolutional models. These methods range from the construction of simple hand-crafted surfaces and volumes (Zhu et al., 2018; 2019; Rebecq et al., 2019), to more complex data-driven techniques (Cannici et al., 2020; Gehrig et al., 2019). This approach achieves state-of-the-art performance, but it discards the sparsity of events, introducing redundant computation. Furthermore, asynchronous event-by-event processing becomes impractical, since all activations of the network would have to be recomputed for each new event. Other architectures are better suited to deal with asynchronous and sparse inputs, like Spiking Neural Networks (SNNs) (O'Connor et al., 2013; Merolla et al., 2014), which are also biologically inspired and hence naturally work with events. Nevertheless SNNs are at a relatively early stage, they still lack a general universal training procedure, and their performance on complex tasks remains limited (Barbier et al., 2021; Rueckauer & Liu, 2018; Lee et al., 2016).

For these reasons, researchers have recently developed methods that leverage the asynchronous and sparse nature of events using traditional deep learning techniques, to favour low latency and lightweight processing while seeking to retain the performance of deep models. Methods based on geometric deep learning have been proposed to model the spatio-temporal structure of events. In order to keep track of temporal context, a history of events is considered, and efficient update mechanisms are devised to avoid the pitfall of recomputation. EventNet (Sekikawa et al., 2019) proposes a recursive variant of PointNet (Qi et al., 2017) for 3D point-cloud processing, which however does not perform hierarchical learning. Similarly, recursive rules that sparsely update the feature maps have been proposed for 2D regular convolutions (Cannici et al., 2019) and sparse convolutions (Messikommer et al., 2020). Graph neural network methods (Li et al., 2021; Schaefer et al., 2022; Mitrokhin et al., 2020) instead represent events as evolving graphs, that are inherently sparse and preserve the temporal structure. We adopt a geometric approach too, since our architecture is based on the convolutional mechanism. However, since it is an RNN model, our method is naturally recursive, instead of requiring to adapt an existing algorithm to work in this way. By making use of the implicit memory of our recurrent units, we do not keep track of a history of events, thus we are not bound to a sliding window of fixed length. Kamal et al. (2023) proposes a transformer architecture that can process batches of events while updating an associative memory. While this performs well in object recognition, we show that our convolutional architecture can be easily applied to dense prediction tasks like object detection, where we obtain the best results.

The combination of CNNs and RNNs has been introduced in ConvLSTM (Shi et al., 2015) to deal with spatio-temporal data, but it can only be applied to dense tensors. Matrix-LSTM (Cannici et al., 2020) then used this concept and adapted it for event streams. While their method preserves sparsity in the computation, it uses a single recurrent layer to build end-to-end differentiable, synchronous and dense representations that are processed by CNNs. We take this approach further, and propose a convolutional RNN, based on the LSTM, that can process events asynchronously in multi-layered hierarchical networks, without the need of a dense CNN. It has been shown that RNNs share many similarities with SNNs (He et al., 2020; Wozniak et al., 2020), hinting that this approach could emerge as the most suited for the domain of event data.

## 3 METHOD

### 3.1 ASYNCHRONOUS EVENT PROCESSING

Event cameras have independent pixels that measure the logarithmic light intensity signal and trigger an event as soon as the change exceeds a certain threshold. Each event encodes the coordinates of the originating pixel $u_i \doteq (x_i, y_i)^\top$, the timestamp of generation $\tau_i$, and the polarity $p_i \in \{-1, +1\}$ that represents the sign of the variation. At any time $T$, the events produced by the camera since initialization are expressed as an ordered sequence of tuples containing these elements. More generally, we can consider a sequence where each tuple stores an arbitrary set of features $v_i$ instead of the polarity only, *i.e.*, the sequence $\mathcal{E} = \{(u_i, \tau_i, v_i) \mid \tau_i \leq T\}$. By processing events asynchronously, our algorithm will produce, by the same time $T$, a sequence of predictions that are continuously updated. Thus, we wish to learn a mapping from a stream of inputs to a stream of output features, or target values (Fig. 1). We do it using a convolutional RNN, which processes the events in the input stream sequentially, capturing the spatial structure due to its convolutional nature, and integrating features in time due to its recurrent nature.

### 3.2 FARSE-CONVOLUTION

We now define the FARSE-Convolution, which acts as the main computational component in our architecture. Inspired by Cannici et al. (2020), we form a 2D grid of recurrent cells, where each cell operates on a spatial neighborhood in the underlying structure of the event stream. This can be either a single pixel or a larger receptive field, and different strides can be used. That is, considering a receptive field centered at $u_c \doteq (x_c, y_c)^\top$, and defining $\mathcal{K}$ the set of positions inside the receptive field, the cell receives as input the event features $v_i$ for the stream $\mathcal{E}^{u_c} = \{(u_i, \tau_i, v_i) \mid \exists k \in \mathcal{K}, u_c + k = u_i\} \subseteq \mathcal{E}$. As in the conventional convolution operation, the same parameters are shared across cells.

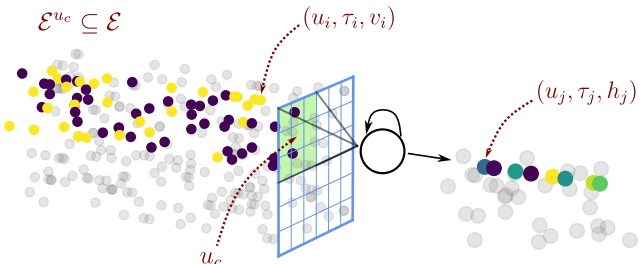

Figure 2: **FARSE-Convolution**. Cells of FARSE-Convolution sequentially process the events of each spatial neighborhood, producing an asynchronous stream of output events.

The recurrent cells are based on the common equations of the LSTM (Hochreiter & Schmidhuber, 1997), but to express spatial kernels in the input-to-state transition, we define different input weight matrices for each position in the receptive field, *i.e.* $W_{ii}^k, W_{if}^k, W_{ig}^k, W_{io}^k, \forall k \in \mathcal{K}$, and sum over all positions that receive simultaneous inputs. Hence, the cell is defined by the set of equations:

$$i_t = \sigma(\sum_{k \in \mathcal{K}} W_{ii}^k v_t^k + b_{ii} + W_{hi}h_{t-1} + b_{hi}) \qquad f_t = \sigma(\sum_{k \in \mathcal{K}} W_{if}^k v_t^k + b_{if} + W_{hf}h_{t-1} + b_{hf})$$

$$g_t = tanh(\sum_{k \in \mathcal{K}} W_{ig}^k v_t^k + b_{ig} + W_{hg}h_{t-1} + b_{hg}) \quad o_t = \sigma(\sum_{k \in \mathcal{K}} W_{io}^k v_t^k + b_{io} + W_{ho}h_{t-1} + b_{ho})$$

$$c_t = f_t \odot c_{t-1} + i_t \odot g_t \qquad\qquad h_t = o_t \odot tanh(c_t)$$

$$(1)$$

where $i_t, f_t, o_t$ are the input, forget and output gates respectively, $g_t$ is the candidate cell, and $c_t, h_t$ are the cell state and hidden state, all at time $t$. Gates use the sigmoid activation $\sigma$, and $\odot$ is the Hadamard product, or element-wise product. Multiple event features $v_t^k$ associated to the same timestamp are processed in the same time step $t$, distinguished by their position $k$.

In contrast to ConvLSTM (Shi et al., 2015), which applies an LSTM on dense 3D tensors using convolutional connections, the independent cells of the FARSE-Convolution are activated sparsely only when inputs are received. Inside the cell, the position $k$ of each input is used to select the corresponding set of weight matrices, and the summation is performed sparsely only for positions where an event is present. ConvLSTM also implements the state-to-state transition as a convolutional filter with the same kernel and stride as the input-to-state transition. In the FARSE-Convolution, the state-to-state transition is convolutional too, since cells do not share a common state (hence the transition is not fully-connected across the grid), but the kernel is effectively restricted to a size of 1. Because cells in FARSE-Convolution are independent, they also asynchronously propagate outputs as soon as these are computed. A stream of output events is generated, each storing the output event features $h_t$, which are computed from the input features $v_t$ using the spatio-temporal context of previous events inside the receptive field. This output stream of events is also spatially structured, since each output is associated to a location in the grid of cells. Thus, we perform a mapping $\mathcal{M} : \mathcal{E}^{in} \to \mathcal{E}^{out}$, as shown in Fig. 2, and we can chain multiple layers of FARSE-Convolution together, having the output stream of one layer become the input stream of the successive one.

### 3.3 SUBMANIFOLD FARSE-CONVOLUTION

When an input event falls into the receptive field of multiple FARSE-Convolution cells, they all trigger an output event simultaneously. After a few consecutive layers, this can result in almost completely dense feature maps generated by a single event in the original input stream. Indeed, FARSE-Convolution suffers from the same problem of input densification that was observed for regular 2D convolution (Graham, 2014; 2015). To address this issue, we define a variant of FARSE-Convolution inspired by Submanifold Sparse Convolutions (SSC) (Graham et al., 2018), which we name Submanifold FARSE-Convolution. SSCs are not mathematically equivalent to regular convolutions, as they only compute an output if the corresponding site in input is active (different from zero). To make this possible, the convolution must be applied with a stride of 1, and the input must be padded so to preserve the spatial dimensions in the output, as in the so-called *same* padding scheme. Submanifold FARSE-Convolution follows these same rules. In our case, positions that are padded

are simply inactive, and cells that have them in their receptive fields will never receive any events in those positions. Then, when an input event is received, all cells that are affected by it update their internal state as usual, but *only* the cell in the corresponding output site triggers an output event and propagates the features forward, as shown in Fig. 3.

A single layer of Submanifold FARSE-Convolution performs exactly the same computation as a regular FARSE-Convolution, since all cells that receive an input are updated as it normally happens. This is necessary to deal with the asynchronicity of the event stream. If we updated, at time step $t$, only the cells whose corresponding site in input is active then we would have single active cells for most time steps, and information from neighbouring positions would not be incorporated. Conversely, since in Submanifold FARSE-Convolution only cells whose corresponding input site is active propagate the events forward, sparsity is maintained in the output and we are guaranteed that the number of events in $\mathcal{E}^{\text{out}}$ is the same as in $\mathcal{E}^{\text{in}}$. This largely affects the computational cost as it prevents the number of events from exploding through the execution of the network.

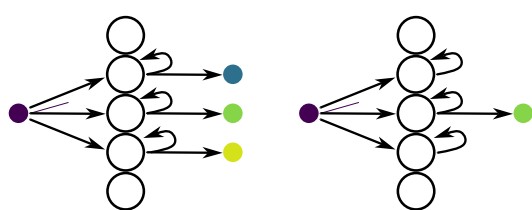

Figure 3: **FARSE-Convolution versus Submanifold FARSE-Convolution**. With FARSE-Convolution (left), all cells that receive the event update their state and trigger an output. With Submanifold FARSE-Convolution (right) instead, all cells update their state, but only the one in site corresponding to the input's triggers an output.

### 3.4 SPARSE POOLING OPERATIONS

Pooling operations are important building blocks in convolutional networks, and they are necessary to downscale the inputs when using Submanifold FARSE-Convolution. Hence, we define sparse variants for pooling operations that can work with event streams. Sparse Pooling operations form a 2D grid of stateless cells. A pooling cell receives input events that fall in its receptive field, and propagates them asynchronously. If multiple events are received simultaneously, their features are aggregated using an aggregation function. If a single event is received, its features are unchanged. Sparse Pooling operations can therefore be interpreted as a special type of FARSE-Convolution where cells are defined by the single non-recurrent equation

$$h_t = \underset{k \in \mathcal{K}}{\alpha}(v_t^k) \tag{2}$$

where $\alpha_{k \in \mathcal{K}}$ is an aggregation function applied over all positions in the receptive field. We consider two pooling operations: Sparse Max Pooling which aggregates features by the element-wise max operation, and Sparse Average Pooling which performs the element-wise average.

### 3.5 TEMPORAL DROPOUT

Pooling operations compress the input spatially. In standard convolutional networks, this allows to learn hierarchically in space and to reduce computation at deeper layers. Event data, however, live in a spatio-temporal domain, and since events are asynchronous, spatial pooling alone is not effective in compressing the stream of events. For these reasons, we define the Temporal Dropout module to compress the event data along the temporal dimension. Temporal Dropout is applied independently on each coordinate of the input stream. This forms once again a 2D grid of asynchronous cells, with receptive fields restricted to single positions. Temporal Dropout is controlled by a window size parameter $l$, and when cells receive input events, they propagate only the last in every window of $l$ events. That is, each cell keeps a counter which is incremented at every new event as follows

$$c_t = (c_{t-1} + 1) \bmod l \tag{3}$$

and triggers an output event only if $c_t = l - 1$. If an output is triggered at time step $t$, the input event at $t$ is propagated without modifying any of its attributes. If an output is not triggered instead, the input event is simply dropped.

Adding one or multiple Temporal Dropout layers in a multi-layered network realizes a network that learns hierarchically in time, *i.e.*, an HRNN (Sordoni et al., 2015). Layers following a Temporal

Dropout learn to focus on longer-term dependencies that span across many time steps. Layers preceding it, focus on shorter-term dependencies, and during training learn to produce optimal summarized representations of events in the same window. Since windows are defined in terms of number of input events, and not time intervals, the output rate of the Temporal Dropout is activity-driven, and the high temporal resolution of events is in principle preserved. Thanks to Temporal Dropout we can effectively compress the stream of events propagated through the network, which majorly impacts the computational complexity of our model.

### 3.6 COMPUTATIONAL COMPLEXITY

We analyze the computational complexity of our architecture in terms of number of floating point operations that are performed for each input event (FLOPs/ev). This metric is independent from the software and hardware implementation, and allows us to compare our method with others in the literature. FARSE-CNN exploits sparsity because, at any layer, only the cells that are affected by an event in input perform some computation. In the 1D case, and when all layers behave as regular convolutions, the maximum number of sites $a_n$ that can be activated at layer $n$ as a consequence of a single event at the input layer is upper bounded as follows

$$a_n \leq 1 + \sum_{i=1}^{n} \frac{k_i - 1}{\prod_{j=i}^{n} s_j} \tag{4}$$

with $k_i$ denoting here the kernel size and $s_i$ the stride of layer $i$. To extend this formula to submanifold convolutions, we consider $k_i = 1$ always if layer $i$ is a Submanifold FARSE-Convolution layer. A proof of this upper bound is provided in Appendix A. In the following, we use $a_n$ to denote the number of active sites in the usual 2D case; For square kernels, this means that the derived upper bound is simply squared.

Upon receiving an input, a cell of FARSE-Convolution performs the same computation as a standard LSTM cell, except that different weight matrices $W_{ii}^k, W_{if}^k, W_{ig}^k, W_{io}^k$ are used for each position, and the results of matrix-vector multiplications for simultaneous inputs are summed together. A cell can receive at most $\min(k^2, a_{n-1})$ inputs, so each of the 4 gates has to aggregate this same amount of $|h|$ dimensional vectors. For all cells that can be active, we obtain the following bound for the FLOPs/ev performed by a FARSE-Convolution at layer $n$:

$$\text{FLOP}_{\text{FARSE-Conv}}^n \leq a_n \left( 8|h| \cdot \left(|h| + |v| + \frac{1}{2}\right) + 4|h| \cdot \left(\min(k^2, a_{n-1}) - 1\right) \right) \tag{5}$$

Equation (5) also applies to Submanifold FARSE-Convolution because the amount of computation performed by each cell is exactly the same as for FARSE-Convolution, as explained in Sec. 3.3. Pooling cells, instead, only incur the cost of aggregating the features of simultaneous inputs. If the Sparse Pooling layer is used with $k = s$, there is no overlap between receptive fields, and over all cells each input is aggregated at most once. The number of operations performed by a Sparse Pooling at layer $n$ is therefore upper bounded as

$$\text{FLOP}_{\text{SP}}^n \leq \begin{cases} a_n \min(k^2, a_{n-1})\text{FLOP}_\alpha & \text{if } k > s \\ a_{n-1}\text{FLOP}_\alpha & \text{if } k = s \end{cases} \tag{6}$$

where $\text{FLOP}_\alpha$ is the amount of FLOPs performed by the aggregation function $\alpha$ on a pair of inputs. We assume $\text{FLOP}_{\max} \leq |v|$ for the max operation, and $\text{FLOP}_{\text{avg}} \leq 2|v|$ for the average operation.

Finally, the temporal compression performed by Temporal Dropout has a large impact on the cost of all layers following it, since it only propagates a portion of its inputs forward. In our experiments in Sec. 4 we compute the FLOPS/ev counting the effective number of inputs and aggregations performed at each layer. However, in Appendix C we show that it is possible to analyze theoretically the effect of temporal compression in our networks.

## 4 EXPERIMENTS

### 4.1 IMPLEMENTATION AND EXPERIMENTAL SETUP

All the modules described in the previous section have been implemented using PyTorch (Paszke et al., 2019). To implement the FARSE-Convolution we adopt a variation, that restricts its parameter

Table 1: **Comparison with asynchronous methods for object recognition.**

| | N-Caltech101 | | N-Cars | |
| --- | --- | --- | --- | --- |
| | Accuracy ↑ | MFLOP/ev ↓ | Accuracy ↑ | MFLOP/ev ↓ |
| HOTS (Lagorce et al., 2017) | 0.210 | 54.0 | 0.624 | 14.0 |
| HATS (Sironi et al., 2018) | 0.642 | 4.3 | 0.902 | 0.03 |
| YOLE (Cannici et al., 2019) | 0.702 | 3659 | 0.927 | 328.16 |
| AsyNet (Messikommer et al., 2020) | 0.745 | 202 | 0.944 | 21.5 |
| NVS-S (Li et al., 2021) | 0.670 | 7.8 | 0.915 | 5.2 |
| EVS-S (Li et al., 2021) | 0.761 | 11.5 | 0.931 | 6.1 |
| AEGNN (Schaefer et al., 2022) | 0.668 | 7.31 | 0.945 | 0.47 |
| EventFormer (Kamal et al., 2023) | **0.848** | **0.048** | 0.943 | **0.013** |
| FARSE-CNN A (Ours) | | - | **0.949** | 2.056 |
| FARSE-CNN B (Ours) | 0.687 | 0.339 | 0.939 | 0.495 |

space, to make it compatible with PyTorch's highly optimized implementation of the LSTM. Instead of using Eq. (1) directly, we define the positional weight matrices $W^k \in \mathbb{R}^{|h| \times |v|} \ \forall k \in \mathcal{K}$, and pass as input to the LSTM the vector $\tilde{v}_t \in \mathbb{R}^{|h|}$ computed as

$$\tilde{v}_t = \sum_{k \in \mathcal{K}} W^k v_t^k. \tag{7}$$

Thus, we pre-multiply the event features with the corresponding weight matrix, and sum together for simultaneous events. The LSTM cell then multiplies this by gate-specific weight matrices, *e.g.*, for the input gate $W_{ii}\tilde{v}_t = W_{ii} \sum_{k \in \mathcal{K}} W^k v_t^k$, which is equivalent to $\sum_{k \in \mathcal{K}} W_{ii} W^k v_t^k$ due to distributive property. Therefore, this formulation follows Eq. (1), but we are imposing a factorization of $W_{ii}^k$ in $W_{ii} W^k$, and similarly for other gates. Notice that after training we can just multiply these weight matrices together once to obtain an implementation equivalent to our original formulation. At the input layer, we represent events with the polarity and the delay from the previous one at the same site, normalized between 0 and a maximum value $d_{max}$ to avoid numerical issues. When a delay larger than $d_{max}$ is computed, the normalized delay is clamped to 1, encoding the fact that an arbitrarily large amount of time might have passed since the last event at the same site. In our case, $d_{max}$ is set to 100 ms, and the first event received at every site is also assigned this delay.

To train our networks, we use the Lightning framework (Falcon & The PyTorch Lightning team, 2019). Training is done using the Adam optimizer (Kingma & Ba, 2014), and the gradients of multiple batches are accumulated to obtain an effective batch size of 256. The learning rate starts at $10^{-3}$ and follows a cosine annealing schedule with warm restarts every 10 epochs (Loshchilov & Hutter, 2017). In each setting, we run the training once with a hold-out validation set to find the optimal number of epochs, and we use this number to train on the full data.

### 4.2 OBJECT RECOGNITION

We evaluate our method on two common datasets. N-Cars (Sironi et al., 2018) targets urban environment applications, and contains recordings of real world objects. It is composed of 24,029 samples, each one is 100 ms long and can either contain a car or be a background scene. N-Caltech101 (Orchard et al., 2015) is the neuromorphic version of Caltech101 (Fei-Fei et al., 2006), generated by moving an event camera in front of still pictures displayed on a monitor. It contains 8,246 samples, each recorded over 300 ms, which are divided into 101 classes.

Our network configuration, selected by hyper-parameters search on the N-Cars dataset, is composed of 11 Submanifold FARSE-Conv layers alternated by 4 blocks of a Sparse Max Pooling and a Temporal Dropout, after which the stream is pooled into a single coordinate and passed to an LSTM and linear classifier. More details on our configurations are in Appendix B. On N-Cars, we train two variants: network A uses window size of 2, and network B uses window size of 4. Both are trained for 130 epochs. On N-Caltech101 we only use network B. We train it on windows of 100 ms for 130 epochs, and then fine-tune it on the full sequences for 30 epochs. We found that this network configuration, combined with the proposed training scheme, offers a better trade-off between performance and training complexity on this dataset. On both datasets we use data augmentations: event coordinates are flipped horizontally at random and translated to random positions inside a maximum frame size.

Table 2: **Accuracy on DVS128 Gesture**. Compared to two synchronous methods that represent the state of the art for frame-based and point-based approaches.

| | Async. | Acc. ↑ | MFLOP/ev ↓ |
|---|---|---|---|
| TBR + I3D (Innocenti et al., 2021) | ✗ | 0.996 | 38820 |
| VMV-GCN (Xie et al., 2022) | ✗ | 0.975 | 330 |
| FARSE-CNN (Ours) | | | |
| w/o state init | ✓ | 0.807 | 1.905 |
| w/ state init | ✓ | 0.966 | 1.905 |

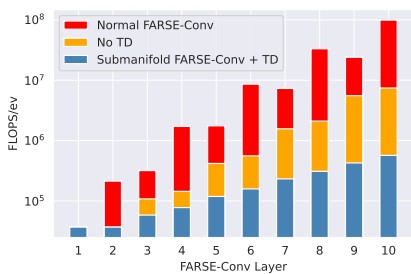

Figure 4: **FLOPS/ev per layer** of network A on N-Cars dataset.

In Tab. 1 we report our results, compared to other asynchronous methods. FARSE-CNN outperforms previous networks on N-Cars when using a window size of 2, while it performs slightly worse, on par with other methods, when the window size is increased to 4, in favor of a 4-times lower complexity. On N-Caltech101, FARSE-CNN achieves among the best accuracy-complexity trade-offs, with only 0.339 MFLOP/ev. While EventFormer surpasses FARSE-CNN on this benchmark, its architecture is limited to high-level tasks, while our solution can easily be applied to dense prediction, as shown in Sec. 4.4. We also emphasize that N-Caltech101 offers a semi-realistic scenario, with static 2D images and identical motion across samples, thus not challenging networks' abilities in exploiting motion-rich information from events. FARSE-CNN on the other hand, excels on real data.

Compared to others, FARSE-CNN is the only method whose complexity does not increase when moving from the N-Cars to the N-Caltech101 dataset. As evident from the analysis of Sec. 3.6, the computation performed by FARSE-CNN does not depend on the spatio-temporal density of events, differently from other methods. In Messikommer et al. (2020) this dependence is formalized through their *fractal dimension*, in graph-based methods Schaefer et al. (2022); Li et al. (2021) the sparsity of the data affects the number of edges in the graph, and in Kamal et al. (2023) the size of the sequences has an impact on the complexity.

Figure 4 shows how the FLOPS/ev are distributed among the layers of FARSE-CNN, considering only Submanifold FARSE-Convolutions with 3×3 kernel, where most of the computation is performed. In the same image we also ablate the complexity of a network that does not use Temporal Dropout and one that uses regular FARSE-Convolutions (with Temporal Dropout), where complexity grows by 1 and 2 orders of magnitude respectively.

### 4.3 GESTURE RECOGNITION

We use the DVS128 Gesture dataset (Amir et al., 2017), which contains 1,342 samples divided in 11 categories of hand and arm gestures. This dataset allows us to study the behavior of our method on long event recordings, in the order of seconds. We use here the same configuration of network A that is used for the N-Cars dataset. Training data is augmented using random coordinates translations.

Since using the full samples is computationally impractical and detrimental to the training itself, we split samples into windows of 100 ms with a stride of 50 ms, and train on those. At inference time, however, FARSE-CNN processes incoming events continuously without re-initialization, and since the model is optimized for sequences of 100 ms, when we feed it the entire recordings its performance degrades quickly. To generalize the model to sequences of indefinite lengths, we adopt the following training strategy. At the start of each epoch we compute and store the state of the entire network after it has processed randomly selected samples for each class (in our case, we store 2 states for each class). Then, during the forward pass, the network is initialized with randomly selected states, or it is left uninitialized with a 20% probability. In this way, the model learns to perform correct inference long after a clean initialization, thus operating flexibly on different time horizons. In both cases, the network is trained for 30 epochs. For a robust evaluation over long sequences, we take the most common prediction (*i.e.*, the statistical mode) over time on the full samples. We increase the accuracy from 80.7% obtained with the simple training to a 96.6% obtained with state initialization. As this is the first asynchronous method that is tested on DVS128 Gesture, we report for reference the accuracy of other *synchronous* methods in comparison with ours in Tab. 2.

## 4.4 OBJECT DETECTION

Event-based object detection meets particular interest in applications where the advantages of event cameras would allow for real time detection in challenging scenarios, such as automotive applications. We use a dataset that targets precisely this field, the Gen1 Automotive dataset (de Tournemire et al., 2020). Gen1 Automotive is collected from almost 40 hours of real world recordings, and contains bounding boxes for 228,123 cars and 27,658 pedestrians. To perform detection with FARSE-CNN, we add to it a fully connected YOLO (Redmon et al., 2016) output layer which detects bounding boxes over a 5×4 grid. The network comprises 8 layers, instead of the 13 that were used for object recognition. We find that making the network deeper in this case does not improve performance, while increasing complexity. This could be explained by the fact that we do not use here any data augmentation, which deeper networks have shown to take more advantage of in our experiments. We make this choice because the detection targets follow a specific distribution in the 2D image space, as explained in de Tournemire et al. (2020), and tampering this distribution with spatial augmentations could be counterproductive.

We optimize a loss that is based on the original YOLO loss, with the small tweaks described here. We use sigmoid activation for all bounding box values, and softmax activation for class scores. For bounding box coordinates we use the mean squared error loss, while for objectness and class prediction we use the cross entropy loss. We set the hyperparameters $\lambda_{coord} = 20$, $\lambda_{noobj} = 1$, and use an additional weight for the objectness error of boxes containing an object $\lambda_{obj} = 0.1$, in order to balance the loss with the new activations. The network is trained and tested on windows of 100 ms preceding each label, and training takes 40 epochs.

Table 3: **Comparison with asynchronous methods on Gen1 Automotive**. For NVS-S we report the result obtained by Schaefer et al. (2022),

|  | mAP ↑ | MFLOP/ev ↓ |
|---|---|---|
| AsyNet Messikommer et al. (2020) | 0.129 | 205 |
| NVS-S Li et al. (2021) | 0.086 | 7.8 |
| AEGNN Schaefer et al. (2022) | 0.163 | 5.26 |
| FARSE-CNN (Ours) | 0.288 | **0.137** |
| FARSE-CNN + NMS (Ours) | **0.300** | **0.137** |

We evaluate our model using the eleven-point mean average precision (mAP) metric (Everingham et al., 2010), implemented in the tool provided by Padilla et al. (2021), as well as the computational complexity. In Tab. 3 we report our results in comparison to other asynchronous methods. FARSE-CNN achieves a 28.8% mAP, which is further improved when we use the non-maximum suppression algorithm (Felzenszwalb et al., 2010) to remove multiple overlapping detections, reaching 30.0% mAP, and outperforming the previous best method, AEGNN, by 13.7%. Considering the individual categories, FARSE-CNN scores 40.5% AP on cars and 19.5% AP on pedestrians, which are under-represented in the dataset. Since this performance is obtained using a relatively shallow network, the complexity is also extremely low, about 38 times lower than the number of operations performed by AEGNN. Thus, our method improves on the state of the art for asynchronous methods on both mAP and complexity for object detection on this dataset. Qualitative results of our network are in the appendices.

## 5 CONCLUSIONS AND FUTURE IMPROVEMENTS

We proposed FARSE-CNN, a novel asynchronous method that leverages sparsity of events in all its components. We showed results of our method on object recognition, and on object detection, where we improved on the state of the art for Gen1 Automotive in both performance and complexity. On gesture recognition we provided results which may be useful as reference for future works.

Our Temporal Dropout compresses features inside windows of events, imposing a simple hierarchy which may not be able to capture the real temporal structure of event data. This could be replaced with a learned latent structure, as in the Multiscale HRNN (Chung et al., 2017). Different RNN models could be employed, like the GRU, to potentially decrease computational complexity even further. Moving towards the neuromorphic approach, more biologically inspired RNN models could be adopted, such as the Spiking Neural Unit proposed in Wozniak et al. (2020) and compared by the authors to the LSTM. Given the connection between RNNs and SNNs, discussed also in He et al. (2020), we believe that our method goes in the direction of bridging the gap between the two lines of research, and hopefully can prove useful also in the study of SNNs applied to event-based vision.

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

## A    NUMBER OF ACTIVE SITES

We now derive and prove the upper bound on the maximum number of sites $a_n$ that can be affected, at layer $n$, by a single event at the input layer, as discussed in Sec. 3.6. Consider first a single layer, that convolves over a 1D space. Generalizing for 2D inputs, the derivations are applied to each dimension independently. The layer can either receive a single event or multiple events in *consecutive* sites. This is because in consecutive applications of convolutional operations, only nearby receptive fields will contain the same sites in input, as long as the convolution is not dilated (which we do not consider here). We can represent these blocks of inputs as small dense frames surrounded by inactive sites, or padding. Then, the number of active sites in output can be computed using the well known formula from convolution arithmetic:

$$o = \left\lfloor \frac{i + 2p - k}{s} \right\rfloor + 1 \tag{8}$$

where $i, o$ are the input and output size respectively, $k, s$ are the kernel size and stride and $p$ is the amount of padding on each side. We want to account for all cells that would contain the input in their receptive field, in a potentially infinite line of cells. This is equivalent to assuming a padding $p = k - 1$, known as the full padding scheme. By substituting this value for the padding, Eq. (8) becomes

$$o = \left\lfloor \frac{i + k - 2}{s} \right\rfloor + 1. \tag{9}$$

When we move to considering multiple layers, the input size of layer $n$ corresponds to the output size of the previous layer, $n - 1$. We can write this as

$$a_n = \left\lfloor \frac{a_{n-1} + k_n - 2}{s_n} \right\rfloor + 1 \tag{10}$$

using $a_n$ for the number of active sites at layer $n$. The base case is $a_0 = 1$, for the single original event in the input layer. We can now solve this recurrence relation to obtain a closed-form upper bound for the number of active sites at layer $n$. To begin with, we upper bound the floor operation $\lfloor . \rfloor$ with its argument, to state

$$a_n \leq \frac{a_{n-1} + k_n - 2}{s_n} + 1. \tag{11}$$

Then, we proceed by substitution:

$$
\begin{aligned}
a_n &\le \frac{a_{n-1} + k_n - 2}{s_n} + 1 \\
&\le \frac{1}{s_n}\left(\frac{a_{n-2} + k_{n-1} - 2}{s_{n-1}} + 1 + k_n - 2\right) + 1 \\
&\le 1 + \frac{k_n - 1}{s_n} + \frac{a_{n-2} + k_{n-1} - 2}{s_n s_{n-1}} \\
&\le 1 + \frac{k_n - 1}{s_n} + \frac{1}{s_n s_{n-1}}\left(\frac{a_{n-3} + k_{n-2} - 2}{s_{n-2}} + 1 + k_{n-1} - 2\right) \\
&\le 1 + \frac{k_n - 1}{s_n} + \frac{k_{n-1} - 1}{s_n s_{n-1}} + \frac{a_{n-3} + k_{n-2} - 2}{s_n s_{n-1} s_{n-2}} \\
&\le 1 + \frac{k_n - 1}{s_n} + \frac{k_{n-1} - 1}{s_n s_{n-1}} + \frac{k_{n-2} - 1}{s_n s_{n-1} s_{n-2}} + \frac{a_{n-4} + k_{n-3} - 2}{s_n s_{n-1} s_{n-2} s_{n-3}}.
\end{aligned}
$$

At this point the pattern is evident, and we can guess our upper bound in closed-form to be the one of Eq. (4), which we repeat here for the reader's convenience:

$$
a_n \le 1 + \sum_{i=1}^{n} \frac{k_i - 1}{\prod_{j=i}^{n} s_j}.
$$

This is proved by induction, assuming that our guess holds for $n - 1$:

$$
\begin{aligned}
a_n &\le \frac{a_{n-1} + k_n - 2}{s_n} + 1 \\
&\le \frac{1}{s_n}\left(1 + \sum_{i=1}^{n-1} \frac{k_i - 1}{\prod_{j=i}^{n-1} s_j} + k_n - 2\right) + 1 \\
&= 1 + \frac{k_n - 1}{s_n} + \frac{1}{s_n}\left(\sum_{i=1}^{n-1} \frac{k_i - 1}{\prod_{j=i}^{n-1} s_j}\right) \\
&= 1 + \frac{k_n - 1}{s_n} + \sum_{i=1}^{n-1} \frac{k_i - 1}{\prod_{j=i}^{n} s_j} \\
&= 1 + \sum_{i=1}^{n} \frac{k_i - 1}{\prod_{j=i}^{n} s_j}
\end{aligned}
$$

which confirms that our guess holds for $n$. The base case $a_0 = 1$ is also trivially proved, because the summation in Eq. (4) is empty for $n < 1$.

Equation (4) holds when all layers are regular convolutional layers, such as FARSE-Convolution and Sparse Pooling layers. When we use Submanifold FARSE-Convolution, instead, we know that the size of the output is the same as the size of the input, *i.e.*, $a_n = a_{n-1}$, and the layer does not influence the number of inputs for layers that follow. Therefore, we simply consider $k_i = 1$ if layer $i$ is a Submanifold FARSE-Convolution, while the stride $s_i$ is already restricted to 1 by design.

## B  NETWORKS DETAILS

In Tab. 4a we show the detailed configurations of the networks used for our experiments on object recognition (Section 4.2) and gesture recognition (Section 4.3). We first apply a $1 \times 1$ Submanifold FARSE-Convolution and then 5 blocks of $3 \times 3$ Submanifold FARSE-Convolutions alternated by compression blocks. In the compression blocks we have a Sparse Max Pooling followed by a Temporal Dropout, so that the network learns to aggregate event features in each spatio-temporal neighborhood. Around the blocks of Submanifold FARSE-Convolutions we use skip connections, by summing together the event features (this is possible because Submanifold FARSE-Convolutions alone do not change the spatio-temporal structure of the event stream). When we apply network B to the N-Caltech101 dataset, we change the first Sparse Max Pooling layer to a kernel $4 \times 4$, performing a stronger spatial compression due to the larger resolution of this dataset.

Table 4: **FARSE-CNN Network Configurations**. Submanifold FARSE-Convolution layers are followed by their kernel and hidden sizes. Sparse MaxPool layers are followed by their kernel size, and the stride for these layers is always equal to the kernel size. TD denotes Temporal Dropout, followed by the window size. Highlighted rows indicate skip connections around the blocks. *The first Sparse MaxPool has kernel size 2 for the N-Cars and DVS128 Gesture datasets, and size 4 for the N-Caltech101 dataset. †The last fully connected layer outputs a flat tensor, but we reshape it into a $5 \times 4$ grid of bounding box predictions according to YOLO.

(a) Configurations A and B, shown in columns.

| FARSE-CNN Configurations | |
|---|---|
| A | B |
| Sub FARSE-Conv 1×1, 16 | |
| Sub FARSE-Conv 3×3, 16 | |
| Sub FARSE-Conv 3×3, 16 | |
| Sparse MaxPool 2×2 / 4×4* | |
| TD 2 | TD 4 |
| Sub FARSE-Conv 3×3, 32 | |
| Sub FARSE-Conv 3×3, 32 | |
| Sparse MaxPool 2×2 | |
| TD 2 | TD 4 |
| Sub FARSE-Conv 3×3, 64 | |
| Sub FARSE-Conv 3×3, 64 | |
| Sparse MaxPool 2×2 | |
| TD 2 | TD 4 |
| Sub FARSE-Conv 3×3, 128 | |
| Sub FARSE-Conv 3×3, 128 | |
| Sparse MaxPool 2×2 | |
| TD 2 | TD 4 |
| Sub FARSE-Conv 3×3, 256 | |
| Sub FARSE-Conv 3×3, 256 | |
| Sparse AvgPool to 1×1 | |
| LSTM, 128 | |
| FC + softmax | |

(b) YOLO configuration for object detection.

| FARSE-CNN YOLO | | | |
|---|---|---|---|
| Layer | | | Output Size |
| Sub FARSE-Conv | 1x1, | 16 | 304×240 |
| Sub FARSE-Conv | 3×3, | 16 | 304×240 |
| Sparse MaxPool | 4×4 | | 76×60 |
| TD | 4 | | 76×60 |
| Sub FARSE-Conv | 3×3, | 32 | 76×60 |
| Sparse MaxPool | 2×2 | | 38×30 |
| TD | 4 | | 38×30 |
| Sub FARSE-Conv | 3×3, | 64 | 38×30 |
| Sparse MaxPool | 2×2 | | 19×15 |
| TD | 4 | | 19×15 |
| Sub FARSE-Conv | 3×3, | 128 | 19×15 |
| Sparse MaxPool | 2×2 | | 10×8 |
| TD | 4 | | 10×8 |
| Sub FARSE-Conv | 3×3, | 256 | 10×8 |
| Sparse MaxPool | 2×2 | | 5×4 |
| FARSE-Conv | 5×4, | 256 | 1 |
| FC | 240 | | 5×4† |

For experiments on object detection (Section 4.4), we build a FARSE-CNN with a fully-connected YOLO output layer, as shown in Tab. 4b. In this case, the blocks of two Submanifold FARSE-Convolutions are replaced with single layers, leading to lower computational complexity as discussed in Sec. 4.4. Skip connections are still used. After the stream is brought down to size $5 \times 4$, a cell of FARSE-Convolution is applied in a fully-connected way (*i.e.*, kernel size and stride are both $5 \times 4$), and its output is passed to a linear fully-connected layer with size 240. The final output is reshaped again into a $5 \times 4$ grid, where each cell predicts two bounding boxes ($2 * 5$ values) and two logits for the class probabilities.

## C   NETWORKS COMPLEXITY

In Sec. 4 we report the MFLOP/ev on each dataset by counting the effective number of inputs received and aggregations performed at each layer. However, the upper bounds provided in Sec. 3.6 give us the complexity of any layer that is reached by input events. To analytically compute the complexity of an entire network, independently of the data it is used on, we must only take care of the effect of Temporal Dropouts. For networks that use Submanifold FARSE-Convolutions, like in our configurations, this can be done by considering the *amortized* cost of all layers. A Temporal Dropout with window size $l$, after an initialization phase (*i.e.*, if the number of inputs received by the layer is much larger than the size of its grid), only propagates $1/l$ of its inputs forward. Hence, the amortization factor of any layer is equal to the product of the inverse window size $1/l$ of all Temporal Dropouts preceding it. For instance, if a certain layer is preceded in the network by two Temporal Dropouts at $n_1$ and $n_2$, its amortization factor will be equal to $1/(l_{n_1} \cdot l_{n_2})$. Note that this amortized analysis cannot be applied to networks that use normal FARSE-Convolutions together with Temporal Dropouts. This is because FARSE-Conv layers generate blocks of contiguous events, as explained

| Network | Dataset | MFLOP/ev | |
| | | Theoretical | Empirical |
|---|---|---|---|
| A | DVS128 Gesture | 2.042 | 1.905 |
| | N-Cars | | 2.056 |
| B | N-Cars | 0.338 | 0.495 |
| | N-Caltech101 | | 0.339 |
| YOLO | Gen1 Automotive | 0.155 | 0.137 |

Table 5: **Theoretical MFLOP/ev of FARSE-CNN networks**. The theoretical complexity depends only on the configuration of each network, although empirically we observe small variations based on the dataset of application.

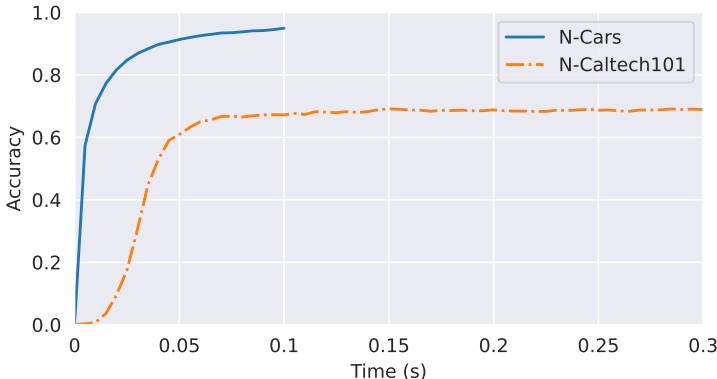

Figure 5: **Accuracy over time** on object recognition datasets. Network A is used for N-Cars, and Network B is used for N-Caltech101.

in Sec. 3.3, some of which are dropped, while others might be propagated due to the independent counters of Temporal Dropout cells. Therefore, computing an upper bound on the complexity of these networks would require a more sophisticated amortized analysis.

Using this reasoning, we obtain for our configurations the values shown in Tab. 5. As can be seen, the empirically measured MFLOP/ev are on some datasets higher than the theoretical ones. This is because the ratio of events that are propagated through the network is higher during an initial phase, and only converges to the one considered in the theoretical amortized analysis as the network receives more input events. Hence, the computed value represents an asymptotic upper bound on the MFLOP/ev of the network. Indeed, in Tab. 5, the empirical complexity remains below the theoretical one for datasets containing longer sequences with more events, such as DVS128 Gesture and Gen1 Automotive.

## D    ACCURACY OVER TIME

FARSE-CNN outputs a stream of predictions driven by incoming events, so we can inspect how the accuracy of the model changes over time. This is visualized in Fig. 5 for object recognition datasets. The plot shows that the model's output has a transient period before converging to a stable answer. On the shorter samples of N-Cars the accuracy steadily increases until the end of the sequences, while on N-Caltech101 it reaches a plateau. In both cases, the model reaches significant accuracy already by the 50 ms mark, long before the total length of samples.

## E    LATENCY

We measure the execution time of the FARSE-CNN YOLO configuration on the Gen1 Automotive dataset. Processing a single event through the entire network takes on average 16.7 ms on an Intel Xeon CPU. Due to the compression along the time dimension of Temporal Dropout, however, not all stages of the network will be executed in response to every input event. In view of this, a

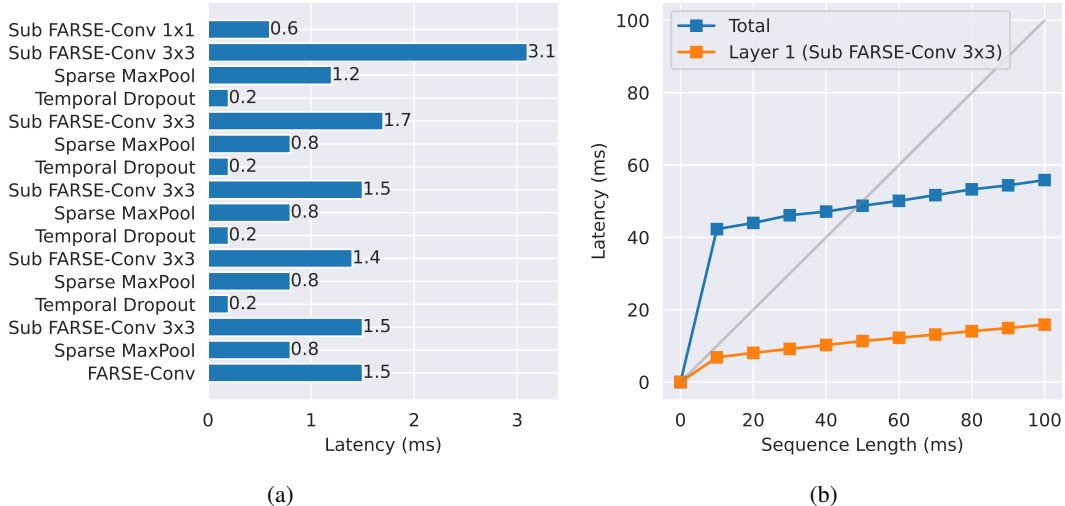

Figure 6: **Execution latency of FARSE-CNN YOLO on Gen1 Automotive**. (a) Latency of individual layers when processing a single event on CPU. (b) Latency over sequences of different time length on GPU. The latency of the slowest layer is also shown separately.

breakdown of the latency for the individual layers is shown in Fig. 6a. As mentioned in Sec. 4.1, our implementation is based on the LSTM provided by PyTorch and it is optimized for parallel processing of multiple sequences on GPU. This allows much faster training times, but it is not as efficient on single step processing. Indeed, processing a single event on an Nvidia Quadro GPU takes 41.6 ms, but the latency scales favorably over longer sequences of events, as shown in Fig. 6b.

We also measure the latency of our method on object recognition datasets following the procedure adopted in Kamal et al. (2023), *i.e.*, measuring the average run time per sample and dividing by the average number of events per sample, on an Nvidia Quadro GPU. We find a latency of 0.0009 ms on N-Caltech101, and 0.0140 ms on N-Cars. For comparison, Kamal et al. (2023) obtained 0.0007 ms on N-Caltech101 and 0.0005 ms on N-Cars using an Nvidia RTX3090. Although their method reports a complexity lower by one order of magnitude on the same datasets, our approach remains competitive in this benchmark on N-Caltech101, where the longer sequences can take advantage of our current parallel implementation.

We believe that large improvements in latency will be obtained through hardware and software implementations optimized for inference, to reduce the GPU overhead observed in Fig. 6b, and to allow the pipelined execution of the architecture.

## F ANALYSIS OF TEMPORAL DROPOUT AND INTERNAL EVENTS

In Fig. 7 we show the ratio of events that are triggered internally by each layer in our networks, on the different datasets. While a small amount of compression is performed by Sparse Pooling operations, the largest impact on the number of internal events is achieved by the temporal compression of Temporal Dropout. When Temporal Dropout uses a window size of 4, less than 1% of the input events received by the network are propagated through to the output. On the N-Caltech101 dataset, we also show additional configurations using larger window size of 6 and 8 to perform even stronger compression. We report the performance of these configurations in Tab. 6, compared to the one we tested

Table 6: **Comparison of Temporal Dropout window size** on N-Caltech101.

| TD Window Size | Accuracy ↑ | MFLOP/ev ↓ |
|---|---|---|
| 4 | 0.687 | 0.339 |
| 6 | 0.678 | 0.190 |
| 8 | 0.660 | 0.148 |

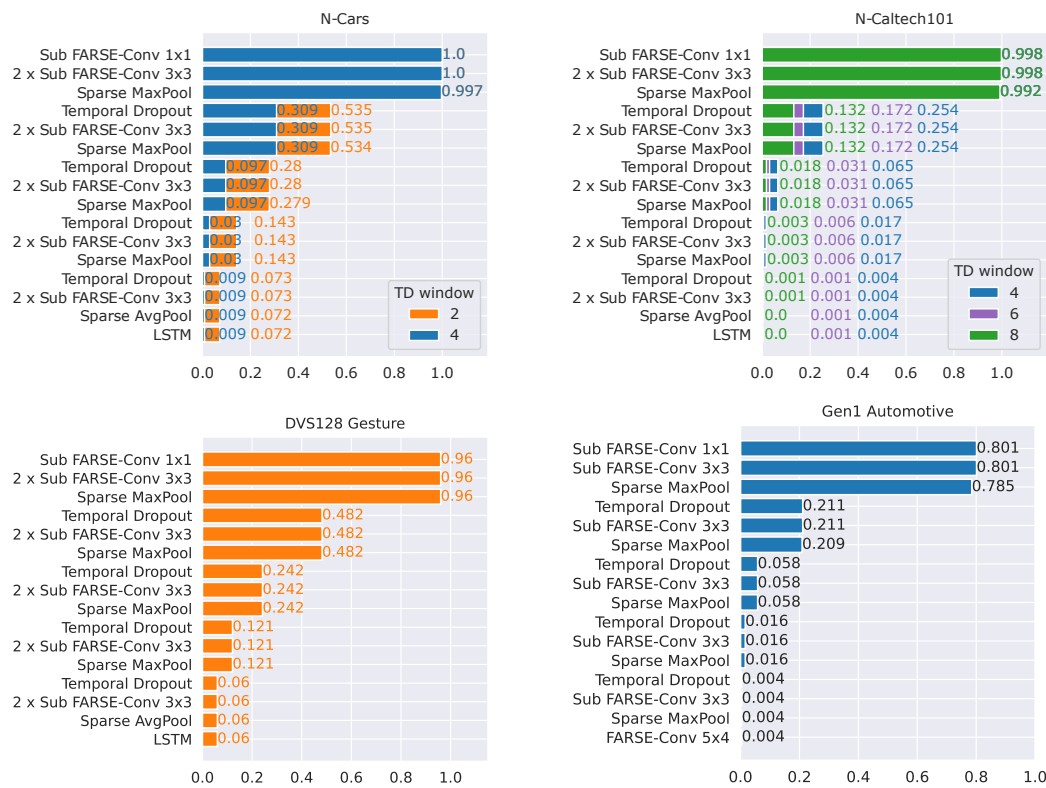

Figure 7: **Internal events triggered at each layer**, expressed as ratio to the number of input events.

in Sec. 4.2. By compromising less than 3% of accuracy, we can reduce the complexity by more than a half.

It can be noted that, in some cases, the ratio of events falls below 1.0 already at the first layers, before any compression module is applied. This is due to the presence of multiple events sharing the same timestamp and coordinates in the original data. Our FARSE-CNN processes them as intended, by aggregating the simultaneous inputs and triggering a single output event, leading to the observed reduction.

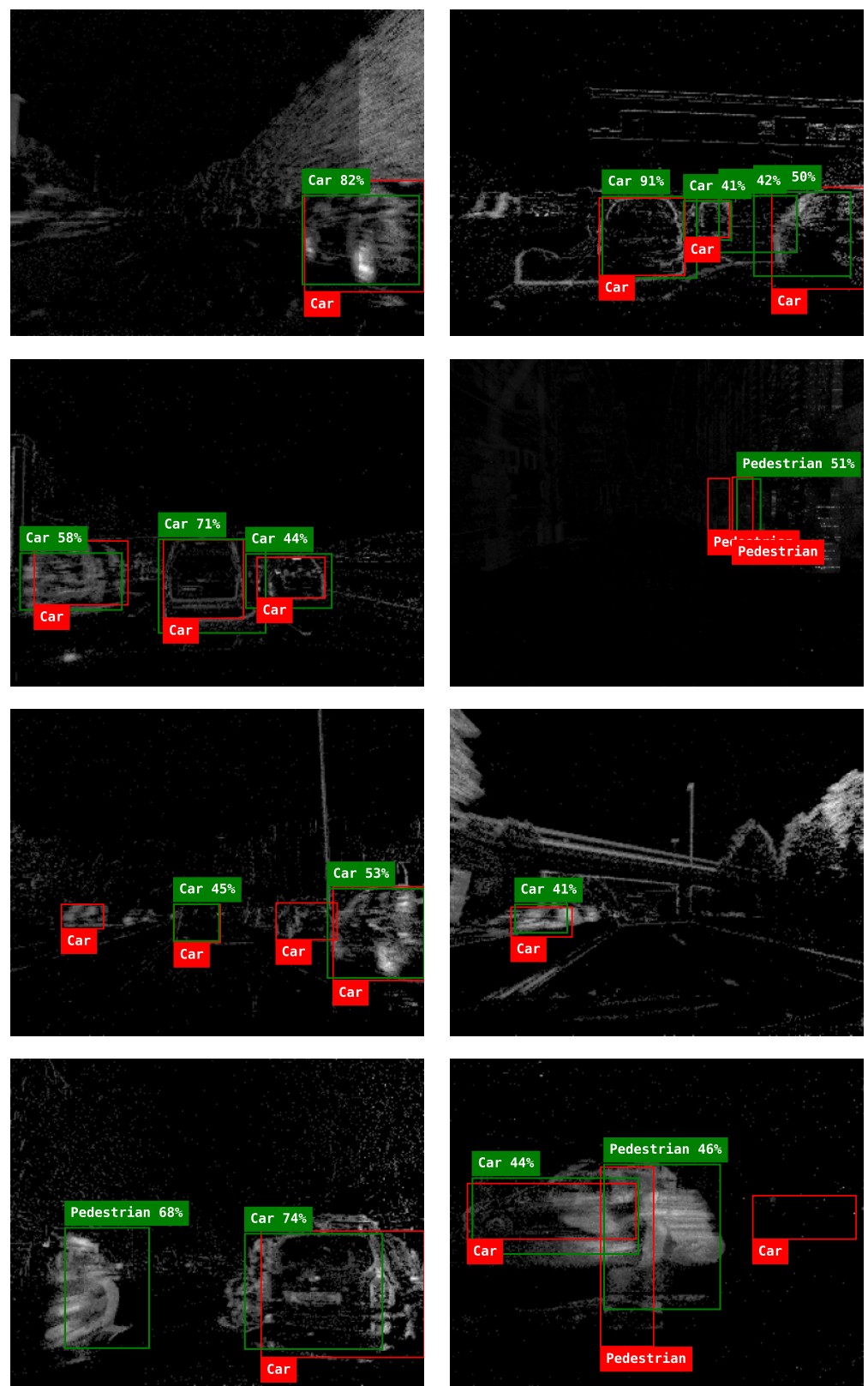

Figure 8: **Examples of FARSE-CNN detections**. Predictions are shown as green boxes, while ground truth boxes are in red. We show both correct predictions and mistakes of our model.

