# OpenReview forum: "FARSE-CNN: Fully Asynchronous, Recurrent and Sparse Event-Based CNN"
_ICLR.cc/2024/Conference — Submitted to ICLR 2024_

### Official Review · Reviewer_WJKv · 2023-10-26

**Soundness:** 3 good
**Presentation:** 3 good
**Contribution:** 3 good
**Rating:** 6
**Confidence:** 3

**Summary:**

The authors propose a model for efficient asynchronous event processing that exploits sparsity. They design the Fully Asynchronous, Recurrent and Sparse Event-Based CNN (FARSE-CNN), a novel multi-layered architecture which combines the mechanisms of recurrent and convolutional neural networks. To build efficient deep networks, they propose compression modules that allow to learn hierarchical features both in space and time.

**Strengths:**

S1: The authors try to present an inherently spiking/event domain based approach for processing asynchronous data from event-like sensors. Often in the related literature you see efforts at converting the data to frame like representations in order to process it using traditional algorithms developed for the synchronous domain. This approach tries to deal with the problem of processing the raw data directly without performing this limiting transformation which could hinder latency. As a result this is the main strength of the paper in my opinion

S2: overall the paper is well written and the algorithm seems interesting, so i think it will be of interest to a subset of the community interested in on the edge based processing approaches

**Weaknesses:**

W1: it is not clear to me if source code will be provided. Please clarify

W2: I would have liked to see a more thorough discussion on the number of events produced internally by this architecture. To run something like this on neuromorphic hardware efficiently, you need to ensure that a sparse number of events are created internally. A discussion on this in the paper, would improve it

**Questions:**

See comments above

---

> ### Author Response · Authors · 2023-11-17
>
> We thank the reviewer for pointing out the necessity of an analysis of the number of internal events. This will be included in the updated manuscript.

---

> > ### Comment · Reviewer_WJKv · 2023-11-21
> >
> > I have gone through the reviewers' and authors' comments. As I indicated in my review, I appreciate the effort of the authors to present an algorithm that does not entail converting events to frames, which is not the case in most papers I see on event based datasets. As a result I have decided to keep my rating unchanged

---

### Official Review · Reviewer_Dqva · 2023-10-30

**Soundness:** 3 good
**Presentation:** 3 good
**Contribution:** 3 good
**Rating:** 5
**Confidence:** 5

**Summary:**

This paper investigates the asynchronous processing of individual event data without converting them into image-like inputs, thereby significantly reducing the overall model energy consumption. The paper designs and implements modules such as FARSE-CNN, SUB-FARSE-CNN, Sparse Pooling, Temporal Dropout, and validates the model's effectiveness in tasks like recognition and detection.

**Strengths:**

1. Based on LSTM design, the paper has implemented a FARSE-CNN for event data, with Sub-FARSE-CNN specifically producing outputs for the central pixel of each cell and updating the state of each cell, addressing the issue of a sharp increase in the number of events after passing through the module.

2. Sparse Pooling compresses event data in the spatial dimension, while Temporal Dropout considers discarding some data in the temporal dimension to encourage the model to learn long-term features, fully utilizing the spatiotemporal characteristics of events.

3. In the tasks of object recognition and object detection, the paper validates the role of the proposed modules in the network, showing their ability to balance computational complexity and accuracy, achieving performance comparable to or better than previous methods. It also achieves performance similar to synchronous methods in gesture recognition tasks.

**Weaknesses:**

1. Although the asynchronous method proposed in the paper handles event data, it does not demonstrate the real-time performance and execution speed of this method. Is there any data available regarding this aspect?

2. For the Temporal Dropout discussed in Contribution 2 and the "l" parameter mentioned in Section 3.5, the experimental section does not provide relevant configurations or discussions.

3. While the paper mentions both FARSE-CNN and SUB-FARSE-CNN, with the latter being an optimized improvement of the former, there is no experimental data to prove the effectiveness of this optimization. For example, there is no performance or computational complexity comparison.

**Questions:**

see Weaknesses

---

> ### Author Response · Authors · 2023-11-17
>
> Regarding the effectiveness of our optimizations, Temporal Dropout and Submanifold FARSE-Convolutions. In Table 1 we have shown that, by changing the window size parameter of Temporal Dropout from 2 to 4, we can reduce complexity by 4 times at the cost of 1% accuracy loss on the NCars dataset. In Figure 4 we also compare the complexity of our network with one that does not use Temporal Dropout, and one that uses normal FARSE-Convolutions instead of Submanifold ones. The complexity growth in those cases (_e.g._, about 100x when using normal FARSE-Convolutions) makes it impractical to train a network due to time and memory constraints since this complexity is directly related to the number of internal events that are generated, hence why we don’t show performance comparisons.

---

### Official Review · Reviewer_bxjc · 2023-10-31

**Soundness:** 2 fair
**Presentation:** 2 fair
**Contribution:** 2 fair
**Rating:** 5
**Confidence:** 4

**Summary:**

A new deep learning architecture - RNN for processing event data.

**Strengths:**

The authors show similar or better performance at higher computation efficiency than other approaches that uses asynchronous methods.

**Weaknesses:**

- How does the complexity of the architecture affect the implementation? Does this architecture of asynchrony give actual speedup?
- Are the datasets shown here sufficient? I am aware of a few other event vision work that looks at some other event data-streams. Can the authors do more SOTA comparisons ?
- Temporal dropout while interesting seems to be an already existing technique? [1] uses some dynamic temporal exit. Further, there are some temporal coding works [2] that use some interesting forms of temporal representation. Can the authors comment on how dropout is different from these?

[1] Li, Yuhang, et al. "SEENN: Towards Temporal Spiking Early-Exit Neural Networks." arXiv preprint arXiv:2304.01230 (2023).
[2] Zhou, Shibo, et al. "Temporal-coded deep spiking neural network with easy training and robust performance." Proceedings of the AAAI conference on artificial intelligence. Vol. 35. No. 12. 2021.

**Questions:**

See above weakness

---

> ### Author Response · Authors · 2023-11-17
>
> In our evaluation, we chose datasets that have been used by previous asynchronous methods to allow for a direct comparison. We believe that additional SOTA comparisons are unfortunately not feasible at this time due to the lack of public code for the baselines. To the best of our knowledge, no public implementation is available for the strongest competitor, EventFormer, as the official project repository is currently empty (https://github.com/udaykamal20/EventFormer), while only the test/inference code is available for AEGNN, and not the training code.
>
> The reviewer noted that there are connections between our Temporal Dropout and previously existing techniques for SNNs. Indeed, the purpose of Temporal Dropout is to introduce a mechanism resemblant of spiking networks in our model, and guarantee that intermediate activations of the network are sparse temporally, as well as spatially. Without this mechanism, our architecture would propagate all events to the output.  As mentioned in our Conclusions, we believe that more sophisticated techniques to exploit temporal structure could be investigated, including Reinforcement Learning approaches as done in SEENN (Li et al., 2023).

---

### Official Review · Reviewer_oYpd · 2023-11-04

**Soundness:** 3 good
**Presentation:** 3 good
**Contribution:** 2 fair
**Rating:** 5
**Confidence:** 4

**Summary:**

The paper intorduces an RNN architecture that is tailored to event-based processing in asynchronous manner. The architecture is evaluated against several other methods on 3 event-based datasets.

**Strengths:**

The paper is written clearly, and the illustrations support the text well. I believe the problem that is being addressed in the paper is important, as the authors mentioned, many modern event-based camera methods rely on image-like representations and thus are suboptiomal for event data processing.

**Weaknesses:**

1) Abstract: It would be better to clarify or paraphrase, as these two sentences seem to contradict each other: "most successful algorithms for event cameras convert batches of events into dense image-like representations" and "achieves similar or better performance than state-of-the-art
asynchronous methods". What was the goal of the paper - to beat the best methods or do develop a sota asynchronous pipeline? I assume the latter, but this needs to be stated more clearly in the abstract.

2) It would help if the evaluation was expanded, since there are not so many event-based datasets available. E.g. CIFAR10-DVS and SL-Animals could be added. A more complex EV-IMO (https://better-flow.github.io/evimo/download_evimo_2.html) could strengthen the paper further.

3) It would be also great to see the performance of the method measured (train / inference separately) on a modern computer or embedded platform. Theoretical computations are valuable, but in practice there are many factors besides flops that can affect the performance. A side-by-side comparison of a few methods would make it more clear to the reader what the implications of the architecture are.

4) From table 1, it seems that the accuracy is not the best (or significantly better compared to competition). The compute cost seems not the lowest as well. I believe a better explanation should be provided to explain the results.

**Questions:**

1) The authors mention, in the introduction, 3D convolutional networks. What is the main difference / advantage of the presented asynchronous scheme compared to 3D cnns, given that both leverage temporal information and, in theory, could be ran event-by-event? An example paper that explores this: https://openaccess.thecvf.com/content_CVPR_2020/papers/Mitrokhin_Learning_Visual_Motion_Segmentation_Using_Event_Surfaces_CVPR_2020_paper.pdf - it would be beneficial to add it to the review section as well.

2) Are there plans to release the source code as a (e.g. Pytorch) package? I believe this would add to the overall contribution of this work.

---

> ### Author Response · Authors · 2023-11-17
>
> About the confusing sentence in the abstract, the goal of the paper is indeed to develop a SOTA asynchronous pipeline. Therefore, the meaning we intended for our claim is "achieves similar or better performance than the state-of-the-art *among* asynchronous methods" *only*.  We will clarify this.
>
> In our evaluation, we focused on datasets used by previous asynchronous methods to allow a comparison with those works. We agree that it will be important to validate our method also on more complex tasks and datasets, such as EV-IMO for motion segmentation and structure from motion. Since this involves non-trivial implementation and would warrant some additional discussion, we left it for future work.
>
> Regarding the performance of our method compared to competitors. It is true that, although our method achieves among the best trade-offs between accuracy and complexity, it is surpassed by EventFormer on some benchmarks. We already acknowledged this in Section 4.4, where we also provide a brief discussion of these results. EventFormer uses a very light, non-hierarchical architecture augmented with an associative memory that is able to achieve good performance on object recognition, particularly so in the semi-realistic scenario of N-Caltech101 that does not present challenging motion-rich information. On the other hand, we emphasize that our FARSE-CNN is easily adapted to the more complex task of object detection on real data, where it achieves the best results in both accuracy and complexity (especially given the revised complexity numbers of AEGNN). Finally, we believe that different methods could prove more effective in different tasks due to their inherent features, and so investigating diverse approaches is beneficial for a field that is still at its early stages, like event vision.
>
> On the differences with 3D convolutional networks. While 3D convolutions can indeed leverage temporal information, standard 3D CNNs require dense inputs, which introduce a lot of redundant computation to the sparse activations of event cameras. For this reason, a line of work focuses on 3D convolutions on sparse graphs that can be updated in event-by-event fashion with less computational burden. The work by Mitrokhin et al. (2020), referenced in the review, also belongs to this family, together with the Graph Neural Network baselines that we considered (NVS-S, EVS-S, AEGNN). We thank the reviewer for pointing us to it, and we will include a reference in the updated manuscript. Differently from these methods, our approach uses a naturally recurrent architecture that can memorize past events in its state, instead of using fixed, explicit time windows. A similar observation, on the difference with LSTM-like approaches, is also made in Mitrokhin et al. (2020) in Section 6.

---

### Author Response · Authors · 2023-11-17

We thank all the reviewers for their comments. They agree that the problem of processing raw event data directly is an important one to address [oYpd, WJKv] and that, in this regard, our approach shows a favorable balance between accuracy and computational complexity [bxjc, Dqva]. Some of our design choices, as well as the clarity of presentation of the paper, were also appreciated.
We address in this comment some common and general questions of the reviewers, while we address specific points in the comments to the individual reviews.

As most of the reviewers have requested, we confirm that the source code of the project *will* be released. Our code based on Pytorch, which can be found already in the supplementary material, is reflective of the final version that will be made public.
We are currently collecting some additional experimental data and analysis requested by the reviewers, and we will soon update the manuscript and our reply. In doing so, we are also including revised numbers for the computational complexity of method AEGNN, as it has come to our attention that the authors of that method have linked an updated version of their paper in the official project repository (https://rpg.ifi.uzh.ch/docs/CVPR22_Schaefer.pdf). Please notice that their numbers have been revised for the worse, making our comparison with one of the most competitive baselines more favorable to our own approach.

---

### Comment · Area_Chair_3hYk · 2023-11-19
**Please engage in reviewer-author discussion**

Dear reviewers,

The paper got diverging scores. The authors have provided their response to the comments. Could you look through their response and other reviews and engage into the discussion with authors? See if their response changes your assessment of the submission?

Thanks!
AC

---

### Author Response · Authors · 2023-11-23

Dear reviewers, we have updated our submission.
The updated manuscript contains two additional appendices, in which we included additional requested experimental data.

Appendix E contains **timing measurements** of our method, both on CPU and on GPU. We provide plots that show the time required by each stage of our pipeline, and the latency scaling over event sequences of different lengths. We believe this section can address questions raised by reviewers oYpd, bxjc and Dqva.

Appendix F contains an **analysis of the internal events** produced by our architecture. We do this for all the network configurations used in the paper, over all the datasets considered. Since the number of internal events is mainly impacted by Temporal Dropout, we include in this analysis additional **configurations for the window size parameter "l"** on the N-Caltech101, that were not used for the experiments in the main text. For this reason, we also report the accuracy and complexity of these networks. We believe this section can address questions raised by reviewers Dqva and WJKv.

In addition to this, we updated the complexity numbers for AEGNN as previously mentioned, clarified a sentence in the abstract and included a reference to [1] (reviewer oYpd).

[1] A. Mitrokhin, Z. Hua, C. Fermüller and Y. Aloimonos, "Learning Visual Motion Segmentation Using Event Surfaces," 2020 IEEE/CVF Conference on Computer Vision and Pattern Recognition (CVPR), Seattle, WA, USA, 2020, pp. 14402-14411, doi: 10.1109/CVPR42600.2020.01442.

---

### Meta-Review · Area_Chair_3hYk · 2023-12-08

**Metareview:**

The paper presents FARSE-CNN, a Fully Asynchronous Recurrent and Sparse Event-Based CNN designed for efficient processing of asynchronous and spatially sparse event data by combining rnn and cnn. FARSE-CNN achieves comparable performance with SOTA asynchronous methods with low computational complexity. Its effectiveness is validated to some extent on the tasks of object recognition, object detection and gesture recognition.

Strengths
The design of joint convolutional and recurrent neural network architectures specifically tailored for asynchronous and sparse event processing​​.
Similar or better performance with higher computation efficiency compared to asynchronous methods.

Weaknesses
No inference computation and speed is reported in the work.
The explanation and validation about effectiveness of the proposed SUB-FARSE-CNN is not clear.
The explanation about the temporal dropout and how the model deal with the case of dense event and various conditions is not clear.

**Justification For Why Not Higher Score:**

No inference computation and speed is reported in the work.
The explanation and validation about effectiveness of the proposed SUB-FARSE-CNN is not clear.
The explanation about the temporal dropout and how the model deal with the case of dense event and various conditions is not clear.

**Justification For Why Not Lower Score:**

This work is recommended as reject.

---

### Decision · Program_Chairs · 2024-01-16

Reject